# Regulation Awareness and Experience of Additional Monitoring among Healthcare Professionals in Finland

**DOI:** 10.3390/healthcare9111540

**Published:** 2021-11-11

**Authors:** Andreas Sandberg, Pauliina Ehlers, Saku Torvinen, Heli Sandberg, Mia Sivén

**Affiliations:** 1Division of Pharmaceutical Chemistry and Technology, Faculty of Pharmacy, University of Helsinki, FI-00014 Helsinki, Finland; pauliina.ehlers@helsinki.fi (P.E.); heli.sandberg@helsinki.fi (H.S.); mia.siven@helsinki.fi (M.S.); 2MedEngine Oy, FI-00130 Helsinki, Finland; saku.torvinen@medengine.fi

**Keywords:** additional monitoring, black triangle, adverse event reporting, pharmacovigilance

## Abstract

Background: Challenges in post-marketing adverse event reporting are generally recognized. To enhance reporting, the concept of additional monitoring was introduced in 2012. Additional monitoring aims to enhance reporting of adverse events (AE) for medicines for which the clinical evidence base is less well developed. Purpose: The purpose was to get a deeper understanding of the underlying reasons why additional monitoring has not increased AE reporting as much as initially hoped. We examined how healthcare professionals (HCPs) in Finland perceive additional monitoring, why they do or do not report AEs more readily for these medicines and how they interact with patients treated with additionally monitored medicines. Methods: An anonymous, open questionnaire was developed and made available online at the e-form portal of University of Helsinki. Physicians, nurses, and pharmacists were invited to complete the questionnaire via their respective trade or area unions. Content analysis of answers to open-ended questions was performed by two independent coders. Results: Pharmacists have the best understanding about additional monitoring but at the same time do not recognize their role in enhancing monitoring. Only 40% of HCPs working with patients knows always or often if a specific medicine is additionally monitored. Half (53%) of HCPs do not tell or tell only rarely patients about additional monitoring. 18% of HCPs reported having received additional monitoring training whereas 29% had received general AE reporting training. AE reporting was more common among HCPs who had received training. Conclusions: Additional monitoring awareness among HCPs and patients should be increased by organizing regular educational events and making additional monitoring more visible. Educational events should emphasize the significance additional monitoring has on patient safety and promote a reporting culture among HCPs.

## 1. Introduction

The early post-marketing period is especially important for establishing a more comprehensive safety profile for new active pharmaceutical ingredients (API), which usually only have safety data about a restricted patient population in controlled experimental conditions [1,2,3]. Spontaneous adverse event (AE) reporting is the main source of safety information during this period although under-reporting of AEs is generally recognized [4].

To enhance AE reporting, the current EU pharmacovigilance (PV) legislation which came into effect in July 2012 introduced the concept of additional monitoring [5,6]. Medicines under additional monitoring have an inverted black triangle (▼) displayed in their package leaflet (PL) and summary of product characteristics (SmPC), together with a statement explaining what the triangle means. The aim of the triangle is to point out to healthcare professionals (HCP) and patients the medicines whose safety is particularly closely monitored by the regulatory authorities [1].

AE reporting has been studied extensively in the 21st century [7,8,9,10,11]. There are however only a few studies concerning additional monitoring as the concept is under a decade old. In Ireland, the awareness of the inverted black triangle symbol among HCPs who knew about additional monitoring was high among pharmacists (>86%) but relatively low among physicians (≈35%) and nurses (15%) [12]. Approximately one-fourth of HCPs who knew about additional monitoring were never or rarely aware if additional monitoring applied to the medicines they used in their practice [12]. Nearly 58% of the HCPs working with patients stated that they did not inform or informed patients only rarely about additional monitoring [12].

The European Medicines Agency (EMA) conducted an EU-wide questionnaire study of additional monitoring in 2017 [13,14]. Only 69% of the HCPs answering the questionnaire reported that they had seen the black triangle and the accompanying statement before [13,14]. Some differences were observed among professions as 83% of the pharmacists, 50% of the physicians, and 42% of the nurses had seen the black triangle [13,14]. In this research, it was concluded that 45% of the pharmacists, 35% of the physicians, and 27% of the nurses had an acceptable understanding of the black triangle and additional monitoring concept [13].

The first package leaflets with the black triangle were introduced to the EU market during spring 2013. In 2019, six years after the introduction of the triangle, we conducted this cross-sectional survey of HCPs (i.e., physicians, pharmacists, and nurses) in Finland to get a more detailed understanding about how HCPs perceive additional monitoring, why they do or do not report AEs more readily for these medicines, and how they interact with patients treated with additionally monitored medicines.

In Finland, pharmacists licensed to practice the profession are Bachelors of Science (B.Sc.) in Pharmacy (1st Cycle Degree) or Masters of Science (M.Sc.) in Pharmacy (2nd cycle degree) graduates [15]. Both groups work with patients and have similar responsibilities in the patient interface. AE reporting is voluntary for HCPs in Finland.

## 2. Methods

### 2.1. Questionnaire Design

An anonymous, open questionnaire was developed and made available online at the e-form portal of University of Helsinki. The final wording of the questionnaire was agreed by an expert panel consisting of two physicians, two nurses, and five pharmacists. Members of the expert panel represented the Finnish healthcare system well as it included experts from industry, academia, hospitals, and open healthcare.

The questionnaire consisted of a cover letter including informed consent statement and a total of 26 questions. Nine questions were open-ended. Two of the 17 multiple-choice questions were designed to measure the knowledge of the respondents. A 5-point Likert scale was used in four questions concerning additional monitoring. The questionnaire is presented in the Appendix A of this article.

The face validity of the questionnaire was tested in a small-scale pilot study with five HCPs. Based on the pilot study, small modifications were made to the questionnaire to improve clarity. No problems were observed with the e-form portal. It was estimated that answering the questionnaire would take 10–15 min.

### 2.2. Questionnaire Distribution

A convenience sample was collected by inviting physicians, nurses, and pharmacists to complete the questionnaire via their respective trade or area unions. The invitation and link to the questionnaire was sent to the respondents via email or by including it to a union newsletter. One reminder was sent in order to maximize the amount of responses. An invitation to complete the survey was also added to the HCP restricted front page of Finnish Medical Network (www.fimnet.fi). No honorarium was provided to Finnish Medical Network, unions or the respondents. Answers were collected from May 2019 to December 2019.

### 2.3. Analysis

#### 2.3.1. Statistical Analysis

IBM SPSS Statistics Version 27 (IBM, Armonk, NY, USA) was used to analyse the data. In the two questions that measured the knowledge of the respondents, the average knowledge score for each HCP subgroup was calculated by summing all correct items and dividing by the total number of items. One-way Analysis of Variance (ANOVA) was used to compare mean knowledge scores between HCP subgroups and most categorical variables. Chi-square test for independence was used in two comparisons of categorical variables. A 5% significance level applies in all hypothesis testing. A Bonferroni correction was applied when multiple group comparisons were made. The Bonferroni corrected alpha level was adjusted to 0.0125 when four comparisons were made and to 0.0167 when three comparisons were made.

#### 2.3.2. Content Analysis

Content analysis of answers to open-ended questions was performed by two independent coders. Data-driven coding approach was used [16].

Reclassification of answer categories was performed in situations where the independent coders had initially created differing categories. Reclassification of answer categories was performed in two questions for physicians and M.Sc. pharmacists, in three questions for nurses and in one question for B.Sc. pharmacists. Re-evaluation of answers in a category was performed in situations where the initial coding resulted in a difference above three answers between the two coders. All differences were discussed until a mutual opinion was reached between the coders.

## 3. Results

A total of 241 responses was received. The response rates could not be calculated as the trade and area unions did not reveal the exact number of email and newsletter recipients. There were 240 complete responses. Six responses did not meet the inclusion criteria (i.e., physician, pharmacist, or nurse). A total of 234 responses were analysed (38 physician, 45 nurse, 36 M.Sc. pharmacist, and 115 B.Sc. pharmacist).

### 3.1. Demographics

Most of the HCPs answering the questionnaire were professionally experienced. 69% (*n* = 161) of the respondents had at least 10 years of experience in their profession. Physicians were the most experienced as 92% (*n* = 35) of them had a minimum experience of 10 years.

The primary workplace was a hospital for most physicians (37%, *n* = 14) and nurses (44%, *n* = 20). Retail pharmacy was the primary workplace for most B.Sc. pharmacists (57%, *n* = 66) whereas the pharmaceutical industry was a major employer of M.Sc. pharmacists (33%, *n* = 12). Demographics are summarised in Table 1.

### 3.2. Adverse Event Reporting Experience

Nearly 56% (*n* = 131) of the HCPs responding to the questionnaire had not reported any AEs during their careers and 38% (*n* = 89) of the HCPs had reported an AE one to nine times. Only 6% (*n* = 14) had reported an AE over 10 times.

The majority of nurses (80%, *n* = 36) and B.Sc. pharmacists (58%, *n* = 67) had not reported any AEs. Correspondingly the majority of physicians and M.Sc. pharmacists had reported at least one AE. A statistically significant difference was observed between nurses and other HCP subgroups. A greater proportion of nurses had not reported any AEs compared to physicians (*p* < 0.001, One-way ANOVA with post-hoc Bonferroni correction) and pharmacists (*p* = 0.004, One-way ANOVA with post-hoc Bonferroni correction). The AE reporting experience is presented in Figure 1.

### 3.3. Adverse Event Reporting Knowledge

Pharmacists scored higher knowledge scores than physicians and nurses when asked about general AE reporting. The average knowledge scores for physicians, nurses, M.Sc. pharmacists, and B.Sc. pharmacists were 4.76, 4.60, 6.19, and 5.34, respectively, out of the possible 8.0. The average knowledge score for all HCPs was 5.24. A statistically significant difference was observed between nurses and pharmacists. Nurses have poorer AE reporting knowledge compared to pharmacists (*p* = 0.016, One-way ANOVA with post-hoc Bonferroni correction). Claims used to test AE reporting knowledge and the results are presented in Table 2.

The Finnish HCPs scored the lowest score on questions concerning topics such as pregnancy and off-label use that were introduced to the EU PV legislation in 2012 [17]. Approximately 70% (*n* = 162) of HCPs knew that patients can report AEs themselves and 51% (*n* = 119) knew that HCPs can report AEs also to the marketing authorization holder (MAH).

Up to 56% (*n* = 132) of the HCPs do not feel that they have enough information on how to report AEs. A prominent share of nurses (73%, *n* = 33) and B.Sc. pharmacists (60%, *n* = 69) would want to have more information about AE reporting. 

### 3.4. Additional Monitoring Knowledge

Approximately 87% (*n* = 203) of the HCPs were aware that some medicines were under additional monitoring before answering the questionnaire.

Questions measuring additional monitoring knowledge rendered the same result as the questions measuring general AE reporting knowledge. Pharmacists scored higher knowledge scores than physicians and nurses. The average knowledge scores for physicians, nurses, M.Sc. pharmacists, and B.Sc. pharmacists were 2.68, 2.16, 3.44, and 3.03, respectively out of the possible 4.0. The average knowledge score for all HCPs was 2.87. A statistically significant difference was observed between nurses and pharmacists. Nurses have poorer additional monitoring knowledge compared to pharmacists (*p* < 0.001, One-way ANOVA with post-hoc Bonferroni correction). Claims used to test additional monitoring knowledge and the results are presented in Table 3.

Approximately 78% (*n* = 184) of the HCPs knew that the reason for additional monitoring is that for these medicines safety information has not yet been collected as much as desired. Almost 15% (*n* = 34) of the HCPs could not answer the true–false statements concerning additional monitoring.

### 3.5. Black Triangle Requirement and Noticeability

Overall, 70% (143/203) of the Finnish HCPs who knew about the additional monitoring concept before answering the questionnaire knew that additionally monitored medicines must have a black triangle in the SmPC, PL, and marketing materials. The black triangle requirement was well-known among pharmacists as 86% (118/138) knew about it. Among other professions this requirement was not that familiar as less than half of the physicians (45%, 14/31) and one-third of the nurses (32%, 11/34) knew about the requirement. A statistically significant difference among all responders was observed between pharmacists and other HCP subgroups. A greater proportion of pharmacists knew about the black triangle requirement compared to physicians (*p* < 0.001, One-way ANOVA with post-hoc Bonferroni correction) and nurses (*p* < 0.001, One-way ANOVA with post-hoc Bonferroni correction). Results are presented in Table 4.

One-fourth (26%, 60/234) of the HCPs had never noticed the black triangle. A breakdown between professions revealed that approximately 40% of physicians (15/38) and nurses (18/45) had not noticed the triangle whereas the corresponding numbers where 8% (3/36) and 21% (24/115) for M.Sc. pharmacists and B.Sc. pharmacists, respectively. In all professions, the majority of responders would prefer that the black triangle and information of additional monitoring is available electronically, preferably in the electronic interface they are using in their everyday tasks.

### 3.6. Effect of Additional Monitoring on Daily Work

Out of the 234 HCPs, 185 worked with patients in their current position. 40% (63/157) of HCPs who worked with patients and knew about the additional monitoring concept stated that they knew always, or often which medicines were under additional monitoring. Correspondingly one-fourth (26%, 41/157) of HCPs did not know or did only rarely know if additional monitoring applied to the medicines.

Half (50%, 101/203) of the HCPs who knew about the additional monitoring concept stated that they report AEs more readily for these medicines compared to other medicines. The most common reason for more active reporting was the desire to increase safety information about the medicine. A significant amount of HCPs emphasized that they report all AEs according to same principles regardless of the medicine. There were responders in all professions who admitted that they do not report AEs more readily as they do not recognize these medicines or do not know how to report AEs.

A quarter (27%) of the HCPs who knew about the additional monitoring concept and worked with patients stated to be always or often more cautious with medicines under additional monitoring whereas 17% stated they are never more cautious. Thirty-five percent of the physicians and 45% of the nurses stated being always or often more cautious whereas only 18% of the pharmacists felt the same. For physicians and nurses, the main reason for being more cautious was the low amount of information available about the medicine. Approximately one-fifth of pharmacists (19%) stated that they are never more cautious. The main reason for not exercising additional caution was that they felt that it was the responsibility of the person prescribing the medicine.

Even 53% of the HCPs who worked with patients and knew about the additional monitoring concept do not tell or tell only rarely the patient about additional monitoring. A quarter (26%) tell the patient always or often. Pharmacists are the most reluctant to tell about additional monitoring as 61% stated that they do not tell or tell only rarely. For pharmacists, the most common reason for not telling was that the information of additional monitoring was considered to be harmful or useless to the patient. 44% of physicians, 28% of nurses, and 19% of pharmacists tell always or often the patient about additional monitoring. For physicians, the most common reason for telling the patient was to get the patient involved in the treatment whereas many nurses felt that telling the patient was their duty.

Only 9% (21/234) of the HCPs answering the questionnaire have received additional monitoring related questions from patients. Most of the questions have concerned the meaning of the inverted black triangle and if it were safe to use the medicine.

### 3.7. Enhancing Adverse Event Reporting of APIs under Additional Monitoring

The Finnish HCPs feel that making AE reporting easier and instructions more clear is most important when trying to enhance AE reporting of APIs under additional monitoring. Many hoped for a simple and fast electronic reporting system that was integrated to the programs used by the HCPs in their everyday practice. Especially nurses wished that the responsibilities and guidelines around AE reporting would be more clear so that it would be easier decide who reports the AEs.

The second most important factor in enhancing AE reporting is to increase communication about additional monitoring and reminding HCPs which medicines are additionally monitored. Several HCPs felt that education around additional monitoring should be increased.

### 3.8. Pharmacovigilance Training of HCPs

Sixty-eight out of the 234 (29%) HCPs have received AE reporting training. The percentage drops to 23% (46/202) when HCPs working in the pharmaceutical industry, academia, and government are excluded. Correspondingly 41 out of the 234 (18%) have received training concerning additional monitoring. The percentage drops to 12% (25/202) when HCPs working in the pharmaceutical industry, academia and government, are excluded. No differences in AE reporting or additional monitoring training prevalence were observed between HCP subgroups (Chi-square test for independence). The PV training prevalence among HCP subgroups is presented in Figure 2.

AE reporting is more common among HCPs who have received AE reporting training (*p* = 0.009, Chi-square test for independence). Sixty-eight percent (15/22) of HCPs who had reported five or more AEs during their career had received general AE reporting training. The corresponding percentages were 38% (39/103) for HCPs who had reported at least one AE and 22% (29/131) for HCPs who had not reported any AEs.

## 4. Discussion

Challenges in post-marketing AE reporting are generally recognized. It is estimated that even 94% of all AEs are not reported [4]. Based on previous research, it seems that making AE reporting mandatory by law does not either increase reporting [18]. In 2012, EU introduced additional monitoring to tackle this gap for medicines for which the clinical evidence base is less well developed. Based on an analysis by EMA, it seems however that reporting activity has not increased significantly [14,19]. Our research revealed some of the underlying reasons why additional monitoring has failed to increase AE reporting in the EU as much as initially hoped. Our research is a first-in-kind to study AE reporting knowledge and experience in Finland.

Our results concerning HCP knowledge about additional monitoring and the inverted black triangle are mostly aligned with previously conducted studies. As in the Irish research, our results suggest that among HCPs pharmacists are best aware that authorities are performing additional monitoring and that the black triangle symbolizes this monitoring. Similar to the EU-wide research conducted by EMA, it is evident that pharmacists have the best understanding about what additional monitoring is and why it is performed. The percentage of HCPs who know about the black triangle requirement is slightly higher in Finland compared to Ireland, especially for physicians (45% vs. ~35%) and nurses (32% vs. 15%) [12]. Correspondingly the percentage of HCPs who have never noticed the black triangle is a bit lower in Finland compared to the EU-wide research (26% vs. 29%) [13,14]. The two-year gap between our research and these two other studies might explain the difference as the number of additionally monitored medicines has increased together with opportunities to notice the triangle.

Based on our results, it is clear that additional monitoring and the black triangle must be made more visible to HCPs working with patients. Only 40% of HCPs who know about additional monitoring and work with patients can always or often tell if additional monitoring applies to the medicine they are giving to the patient. Exactly the same result was observed in the Irish research [12]. The list of additionally monitored medicines contains already hundreds of APIs and for HCPs to remember this list by heart is impossible [20]. It is now clear that HCPs prefer to get information about additional monitoring via the electronic interface they use while working with patients. Adding a notification in the interface before prescribing, administering, or dispensing the medicine would make sure the information is at hand when needed.

Half of Finnish HCPs state to report AEs more readily for additionally monitored medicines if they are aware of additional monitoring. It was reassuring to discover that a big portion of these HCPs report AEs because they want to increase knowledge about the medicines and improve patient safety. It is nevertheless troublesome to notice that some HCPs still do not see the value of single AE reports and neglect reporting entirely for example because it is too laborious.

Our research revealed that unlike physicians and nurses, a big part of pharmacists do not see the significance of their efforts in the additional monitoring process. The majority feel that telling the patient about additional monitoring and being more cautious with these treatments is the responsibility of the physician or nurse prescribing the medicine. Whereas for physicians the most common reason for telling the patient about additional monitoring was to get the patient involved in the treatment; the pharmacists refrained from informing the patient. In many instances, pharmacists see the patients much more often than the nurse or physician. With this in mind, we can argue that pharmacists actually have a pivotal role in the process as they can readily disseminate information about additional monitoring and also collect new safety information.

Our results suggest that over half (53%) of HCPs do not tell or tell only rarely the patient about additional monitoring. At the same time, it was made clear that less than 10% of the HCPs have received additional monitoring or black triangle-related questions from patients. Currently the black triangle and the accompanying statement are only present in the SmPC and PL. Previous research suggest that many patients find PLs hard to comprehend and have difficulties in finding the information they are looking for [21,22,23]. Many patients do not either read the PL [21,22,23]. Adding the additional monitoring information also to outer packaging might help in raising awareness especially among the public. This could in turn increase discussion among patients and their HCPs and ultimately enhance AE reporting.

Based on our research, AE reporting of APIs under additional monitoring could be enhanced by increasing awareness among HCPs via frequent information campaigns and active training. A statistical significance between training combined with PV information campaigns and increased AE reporting rate was found in a Portuguese study [24]. Training without information campaigns failed to show statistical significance [24]. Only a fraction of the Finnish HCPs working with patients reported to having received AE reporting or additional monitoring training. It is therefore of utmost importance to get HCPs educated about PV and keep PV on display e.g., by making educational events recurring. Based on our results and previous research, the training should be designed to change erroneous beliefs and promote a reporting culture among HCPs [25].

The majority HCPs answering our questionnaire were professionally experienced as 69% had at least 10 years of experience. This may be a source of bias as HCPs receiving their basic education after additional monitoring implementation are underrepresented in our sample. Correspondingly M.Sc. pharmacists working in the pharmaceutical industry or the government are overrepresented in our sample as nearly 40% of M.Sc. pharmacist responders belonged to this group and are probably better trained on the current legislation and AE reporting guidelines. This effect is however diluted in the whole pharmacist group as only 16% of the pharmacists worked in the industry or the government.

Non-response bias is the main limitation of this research. Response rates could not be calculated due to the method of questionnaire distribution but based on previous research, it is expected that the response rate is low [12]. In this research, HCPs reflected their past actions and knowledge, which is the second most important source of possible bias as answers depended on own recollection and honesty. Although sources of bias must be taken into consideration, our results are well aligned with previous research and with this sample size largely generalisable to other European countries with similar regulation and HCP educational criteria.

## 5. Conclusions

In conclusion, pharmacists were found to be best aware of additional monitoring but at the same time they did not recognize their role in the additional monitoring process. Like other HCPs, pharmacists could also serve an important role in getting the patient involved in treatment and tell them about the importance of additional monitoring. This discovery requires confirmation in future research but it is nevertheless certain that additional monitoring awareness among HCPs working with patients should be increased to avoid any possible misconceptions. Clarification of roles and responsibilities between different healthcare professions should also be emphasized. Locations where many different HCPs are in contact with the same patient (e.g., hospitals) are especially vulnerable and require in-house procedures stating when AEs are reported and by whom. Finally we recommend that health authorities look into the benefits and risks associated with adding the black triangle also to the outer packaging of additionally monitored medicines. We believe that making additional monitoring more visible to HCPs and patients will increase AE reporting and thereby promote patient safety.

## Figures and Tables

**Figure 1 healthcare-09-01540-f001:**
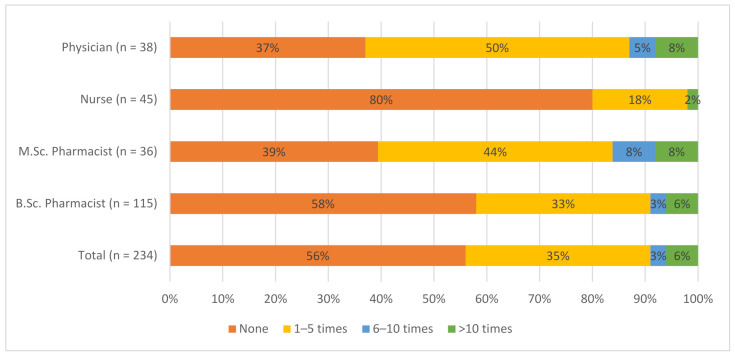
Adverse event reporting experience among healthcare professionals. Questionnaire question: How many times have you reported an adverse event to the local health authority (Fimea) or to the marketing authorization holder?

**Figure 2 healthcare-09-01540-f002:**
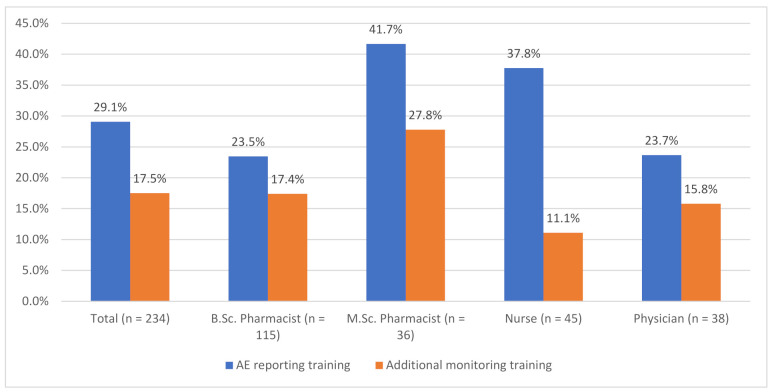
Percentage of healthcare professionals who have received pharmacovigilance training.

**Table 1 healthcare-09-01540-t001:** Healthcare professional demographics.

Profession	Physician	Nurse	M.Sc. Pharmacist	B.Sc. Pharmacist	Total
Group size (*n*)	38	45	36	115	234
Years in practice, % (*n*)					
<5	5.3 (2)	17.8 (8)	19.4 (7)	13.9 (16)	14.1 (33)
5–9	2.6 (1)	15.6 (7)	30.6 (11)	18.3 (21)	17.1 (40)
10–19	18.4 (7)	31.1 (14)	30.6 (11)	29.6 (34)	28.2 (66)
>20	73.7 (28)	35.6 (16)	19.4 (7)	38.3 (44)	40.6 (95)
Primary workplace, % (*n*)					
Hospital	36.8 (14)	44.4 (20)	0 (0)	7.8 (9)	18.4 (43)
Private clinic	23.7 (9)	2.2 (1)	0 (0)	0 (0)	4.3 (10)
Healthcare center	18.4 (7)	20.0 (9)	0 (0)	4.3 (5)	9.0 (21)
Government	5.3 (2)	(0)	5.6 (2)	3.5 (4)	3.4 (8)
Nursing home	2.6 (1)	17.8 (8)	0 (0)	0 (0)	3.8 (9)
Retail pharmacy	0 (0)	0 (0)	36.1 (13)	57.4 (66)	33.8 (79)
Pharmaceutical industry	0 (0)	0 (0)	33.3 (12)	5.2 (6)	7.7 (18)
Hospital pharmacy	0 (0)	0 (0)	13.9 (5)	16.5 (19)	10.3 (24)
Other	13.2 (5)	15.6 (7)	11.1 (4)	5.2 (6)	9.4 (22)

**Table 2 healthcare-09-01540-t002:** Adverse event reporting knowledge among Finnish HCPs, % correct, (*n*/*n*) correct answers/all answers.

Claim ^a^	Physician (*n* = 38)	Nurse (*n* = 45)	M.Sc. Pharmacist (*n* = 36)	B.Sc. Pharmacist (*n* = 115)	Total (*n* = 234)
HCPs are encouraged to report AEs even if uncertain medicine is the culprit (yes)	89.5% (34/38)	57.8% (26/45)	91.7% (33/36)	82.6% (95/115)	80.3% (188/234)
HCPs are encouraged to report AEs even if they do not have all the details of the event (yes)	68.4% (26/38)	60.0% (27/45)	86.1% (31/36)	72.2% (83/115)	71.4% (167/234)
All serious AEs are known once the medicine enters the market (no)	92.1% (35/38)	80.0% (36/45)	97.2% (35/36)	94.8% (109/115)	91.9% (215/234)
AEs reported by Finnish HCPs are handled locally and do not influence safety information in other countries (no)	92.1% (35/38)	80.0% (36/45)	94.4% (34/36)	93.9% (108/115)	91.0% (213/234)
Patients themselves can report AEs to HA or MAHs (yes)	55.3% (21/38)	57.8% (26/45)	86.1% (31/36)	73.0% (84/115)	69.2% (162/234)
HCPs are also encouraged to report overdoses, misuse, and medication errors (yes)	34.2% (13/38)	62.2% (28/45)	63.9% (23/36)	37.4% (43/115)	45.7% (107/234)
HCPs are encouraged to report medicine use during pregnancy (yes)	7.9% (3/38)	17.8% (8/45)	41.7% (15/36)	24.4% (28/115)	23.1% (54/234)
HCPs should report AEs to the local HA, not to MAH (no)	36.8% (14/38)	44.4% (20/45)	58.3% (21/36)	55.7% (64/115)	50.9% (119/234)
I do not know ^b^	7.9% (3/38)	15.6% (7/45)	2.8% (1/36)	5.2% (6/115)	7.3% (17/234)
Total amount of right answers	59.5% (181/304)	57.5% (207/360)	77.4% (223/288)	66.7% (614/920)	65.4% (1225/1872)
Average knowledge score per responder	4.76 (181/38)	4.6 (207/45)	6.19 (223/36)	5.34 (614/115)	5.24 (1225/234)

AE adverse event, HCP healthcare professional, HA health authority, MAH marketing authorization holder. ^a^ Correct answer is presented in brackets after the claim. ^b^ “I do not know” was an option to HCPs who could not answer the claims. The answers of HCPs who chose “I do not know” were considered as wrong answers in the analysis of results.

**Table 3 healthcare-09-01540-t003:** Additional monitoring knowledge among Finnish HCPs: Why are some medicines additionally monitored? % correct, (*n*/*n*) correct answers/all answers.

Claim ^a^	Physician (*n* = 38)	Nurse (*n* = 45)	M.Sc. Pharmacist (*n* = 36)	B.Sc. Pharmacist (*n* = 115)	Total (*n* = 234)
These medicines have more serious AEs (No)	50.0% (19/38)	35.6% (16/45)	69.4% (25/36)	57.4% (66/115)	53.9% (126/234)
AEs are more common with these medicines (No)	68.4% (26/38)	44.4% (20/45)	88.9% (32/36)	77.4% (89/115)	71.4% (167/234)
Safety information has not yet been collected as much as desired for these medicines (Yes)	73.7% (28/38)	60.0% (27/45)	94.4% (34/36)	82.6% (95/115)	78.6% (184/234)
No reason. All medicines will be additionally monitored after the transition period (No)	76.3% (29/38)	75.6% (34/45)	91.7% (33/36)	85.2% (98/115)	82.9% (194/234)
I do not know ^b^	21.1% (8/38)	24.4% (11/45)	5.6% (2/36)	11.3% (13/115)	14.5% (34/234)
Total amount of right answers	67.1% (102/152)	53.9% (97/180)	86.1% (124/144)	75.7% (348/460)	71.7% (671/936)
Average knowledge score per responder	2.68 (102/38)	2.16 (97/45)	3.44 (124/36)	3.03 (348/115)	2.87 (671/234)

AE adverse event, HCP healthcare professional. ^a^ Correct answer is presented in brackets after the claim. ^b^ “I do not know” was an option to HCPs who could not answer the claims. The answers of HCPs who chose “I do not know” were considered as wrong answers in the analysis of results.

**Table 4 healthcare-09-01540-t004:** Percentage of HCPs who knew about the additional monitoring concept and black triangle requirement.

Profession	Knew about Additional Monitoring	Knew about Black Triangle Requirement	Knew about Additional Monitoring and Black Triangle Requirement
Physician	81.6% (31/38)	36.8% (14/38)	45.2% (14/31)
Nurse	75.6% (34/45)	24.4% (11/45)	32.4% (11/34)
M.Sc. pharmacist	97.2% (35/36)	86.1% (31/36)	88.6% (31/35)
B.Sc. pharmacist	89.6% (103/115)	75.7% (87/115)	84.5% (87/103)
Total	86.8% (203/234)	61.1% (143/234)	70.4% (143/203)

HCP healthcare professional.

## Data Availability

Data is not available online.

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
