# Peer review of "Regulation Awareness and Experience of Additional Monitoring among Healthcare Professionals in Finland"

_healthcare, 2021, doi:10.3390/healthcare9111540_

Round 1

Reviewer 1 Report

This paper explores the regulation awareness among Finnish health care professionals regarding "Additional Monitoring"

The authors explored the knowledge level among physicians, nurses and pharmacists. The work is well designed and well presented, however, it can be improved and the presentation can be highly enhanced.

Major comments:

  1. The title is general and does not reflect the content clearly, I suggest replacing it with: "Regulation awareness regarding medicines under "additional monitoring" among health care professionals in Finland" or any other title that emphasizes that the knowledge is about the concept and regulation rather than the process of additional monitoring itself.
  2. Additional monitoring term was used inconsistently, sometimes it refers to the concept, and in other cases to the regulation. I suggest revising the whole manuscript and adding extra clarification. for example Page 7 paragraph 5: "Even 53% of the HCPs who worked with patients and knew about additional monitoring never or rarely tell the patient about additional monitoring." knew about additional monitoring what? procedures? regulations? concept? methods? such terms need more clarification to make the flow of the paper smoother and easier to understand.
  3. Statistical analysis requires clarifications and more details on how p values were calculated. For example, What is the Bonferroni test? is it ANOVA with Bonferroni correction?
  4. Authors claim that pharmacists did not recognize their role in the additional monitoring process without clear justification. I suggest giving more explanation on how this conclusion was made.
  5. Authors claim that "additional monitoring" did not lead to increase of the AE reporting without providing data or reference.

Specific comments:

Abstract 

  • Starting abstract with purpose is strange. Please start with a brief introduction and definition of the problem, then the purpose can follow. 
  • What does additional monitoring really mean, please give a brief explanation? 
  • AE? => Adverse events? HCP => (Health care providers) Please mention the full term for abbreviations at first instance. 
  • “Statistical analysis was performed to analyze the significance of the results” That is obvious. Please provide some details to make the sentence useful or remove it because it is well understood.

Introduction:

  • Some sentences require rephrasing: 
    • Studies concerning additional monitoring are however only a few as the concept is under a decade old.
    • In a EU-wide research conducted by the European Medicines Agency (EMA) in 2017 it was observed that 69% of the HCPs.
  • Undergraduate means “a student at a college or university who has not received a first and especially a bachelor's degree”. Please find a better term to be used here.

Results:

  • 3.2 what does the Bonferroni test mean? It is ANOVA with Bonferroni correction? Please clarify the statistical method used to calculate P values. 
  • 3.3 Bonferroni test; please provide more details and better terms.
  • What does “I do not know” imply? If "I don’t know" was considered a wrong answer (lack of knowledge), please clarify in the text.
  • 3.4 Bonferroni test, please clarify.
  • 3.5 Please clarify what does Bonferroni test mean. 

Author Response

Response to Reviewers’ Comments 

We are very thankful for your comments and suggestions for improvement of the manuscript, as well as for your positive feedback. We have answered your comments or suggestions in a point-by-point manner (below) and marked the changes with “track changes” function in the revised manuscript.

Reviewer 1

Responses to Major comments:

  1. Thank you for this comment. Title is revised so that it emphasizes that the knowledge is about the concept and regulation rather than the process of additional monitoring itself.
  2. The terminology is clarified through the manuscript. As an example, the sentence in section 3.6 is revised as “Even 53% of the HCPs who worked with patients and knew about the additional monitoring concept do not tell or tell only rarely the patient about additional monitoring.”. (Previous “Even 53% of the HCPs who worked with patients and knew about additional monitoring never or rarely tell the patient about additional monitoring.”)
  3. Statistical analysis has been described in more detail in methods and in the text.
  4. Better justification is given for the claim that pharmacists seem not to recognize their role in additional monitoring. Discussion (paragraph 5) “Our research revealed that unlike physicians and nurses a big part of pharmacists do not see the significance of their efforts in the additional monitoring process. The majority feel that telling the patient about additional monitoring and being more cautious with these treatments is the responsibility of the physician or nurse prescribing the medicine. Whereas for physicians the most common reason for telling the patient about additional monitoring was to get the patient involved in the treatment, the pharmacists refrained from informing the patient.”. Discussion (paragraph 10) “In conclusion, pharmacists were found to be best aware of additional monitoring but at the same time they did not recognize their role in the additional monitoring process. Like other HCPs, pharmacists could also serve an important role in getting the patient involved in treatment and tell them about the importance of additional monitoring.
  5. Thank you for the comment. References are provided in Discussion (paragraph 1): “Based on an analysis by EMA it seems however that reporting activity has not increased significantly.14,19

Refs:

  1. European Medicines Agency (EMA). European Medicines Agency and Member States joint report to the European Commission on the experience with the list of products subject to additional monitoring. 2018; EMA/385597/2019. https://www.ema.europa.eu/en/documents/report/european-medicines-agency-member-states-joint-report-european-commission-experience-list-products_en.pdf
  2. Segec A, Slattery J, Morales DR, Januskiene J, Kurz X, Arlett P. Does additional monitoring status increase the reporting of adverse drug reactions? An interrupted time series analysis of EudraVigilance data. Pharmacoepidemiol Drug Saf. 2021;30:350-359. https://doi.org/10.1002/pds.5174

Responses on Specific comments:

Abstract: Section “Background” is added.  “Background: Challenges in post-marketing adverse event reporting are generally recognized. To enhance reporting, the concept of additional monitoring has been introduced in 2012.” Also, a brief explanation for additional monitoring is added, as suggested.

Abstract: Full term for abbreviations (AE, HCP) are given at first instance.

Abstract: Sentence “Statistical analysis was performed to analyse the significance” is removed, as recommended by the reviewer.

Introduction (Rephrasing sentences):

Thank you for this comment. The sentence is rephrased as follows “There are however only a few studies concerning additional monitoring as the concept is under a decade old.” (Introduction, paragraph 3)

The sentence “In a EU-wide research…” is clarified as follows in the revised manuscript “The European Medicines Agency (EMA) conducted an EU-wide questionnaire study of additional monitoring in 2017. Only 69% of the HCPs answering the questionnaire reported that they had seen the black triangle and the accompanying statement before.” (Introduction, paragraph 4)

The text has been revised regards the term “undergraduate” as follows “In Finland, pharmacists licenced to practise the profession are Bachelors of Science in Pharmacy (1st Cycle Degree) or Masters of Science in Pharmacy ( 2nd cycle degree) graduates” (ref. Hirvonen et al. 2019).

Reference added  (15. Hirvonen, J., Salminen, O., Vuorensola, K., Katajavuori, N., Huhtala, H., Atkinson, J., 2019. Pharmacy practice and education in Finland. Pharmacy 7 (1), 21. https://doi.org/10.3390/pharmacy7010021)

Results:

Statistical analysis is now described in more detail in section 2.3.1: “A Bonferroni correction was applied when multiple group comparisons were made. The Bonferroni corrected alpha level was adjusted to 0.0125 when four comparisons were made and to 0.0167 when three comparisons were made.” Also, Bonferroni correction is clarified in sections 3.2; 3.3; 3.4 and 3.5, as suggested.

Thank you for the important comment on “What does “I do not know” imply?” The answer category has been clarified in the subheading in Tables 1 and 2 in the revised manuscript: “”I do not know” was an option to HCPs who could not answer the claims. The answers of HCPs who chose ”I do not know” were considered as wrong answers in the analysis of results.

Reviewer 2 Report

A well put together paper that has importance for health practitioners and consumers. However, there are some minor grammatical errors that need correcting - for example:

In the abstract  - extra words - "18% of HCPs reported to having"

Incomplete sentences: -  under table 2 - "Namely questions concerning reporting ...."; 3rd paragraph in section 3.6 - "Approximately one fifth of pharmacists (19%) are never more cautious".  

I suspect the issue with some sentences, like the last example above, are related to using the survey responses as part of the sentence without highlighting that 'never' is one of the question options for a response. This occurs in a number of places making the grammar a bit clumsy (so to speak). This is why I have ticked 'moderate' with respect to the English language and style question above as these are not typographical errors but minor grammar errors. Please check the paper throughout.

Institutional ethical review has not been referred to within the paper and should be included.

Author Response

Response to Reviewers’ Comments 

We are very thankful for your comments and suggestions for improvement of the manuscript, as well as for your positive feedback. We have answered your comments or suggestions in a point-by-point manner (below) and marked the changes with “track changes” function in the revised manuscript.

Reviewer 2 

Thank you for commenting on grammatical issues and the clarifying examples. It helped us in revisions towards more concise and clear manuscript. The manuscript is revised accordingly.

Ethical review is now described more specifically in the revised manuscript (Section Ethics Statement): "The research was approved by the University of Helsinki Ethical Review Board in the Humanities and Social and Behavioural Sciences on the 25th of April 2019 (statement 22/2019). No other institutional statements were needed.” Reference Finnish National Board on Research Integrity, 2019 

Reference added (26. Finnish National Board on Research Integrity TENK 3/2019 (page  

https://tenk.fi/sites/default/files/2021-01/Ethical_review_in_human_sciences_2020.pdf)

Reviewer 3 Report

This is a well written paper with some interesting results that will contribute to the overall knowledge on this topic.  Some of the results are not surprising, but the overall paper is interesting and novel.
Overall, I found few errors, issues or problems.  I congratulate you on this.  I have one comment, that can be taken on board or not.  It does not affect the paper in any way.  Most of these service evaluations identify issues with knowledge on the topic.  Then education programmes are always the answer.  However, education on a topic will only last for long and if you re-evaluate their knowledge level again in 6 to 12 months, it has not really improved.  Can something else be done to make sure that the knowledge level increases or at least plateaus at a higher level than before education?  I feel an education programme is always a quick fix.  How about compulsory  training or continuous training to keep the message ongoing and alive.

Author Response

Response to Reviewers’ Comments 

We are very thankful for your comments and suggestions for improvement of the manuscript, as well as for your positive feedback. We have answered your comments or suggestions in a point-by-point manner (below) and marked the changes with “track changes” function in the revised manuscript.

Reviewer 3

Thank you for bringing up this important comment relating to education. In revised manuscript, we have elaborated the issue more thoroughly. In Discussion (paragraph 7) we now highlight the importance of continuous education in proficiency enhancement of professionals.